# Zero-Shot Learning Through Cross-Modal Transfer

**Richard Socher, Milind Ganjoo, Christopher D. Manning, Andrew Y. Ng**
Computer Science Department, Stanford University, Stanford, CA 94305, USA
richard@socher.org, {mganjoo, manning}@stanford.edu, ang@cs.stanford.edu

## Abstract

This work introduces a model that can recognize objects in images even if no training data is available for the object class. The only necessary knowledge about unseen visual categories comes from unsupervised text corpora. Unlike previous zero-shot learning models, which can only differentiate between unseen classes, our model can operate on a mixture of seen and unseen classes, simultaneously obtaining state of the art performance on classes with thousands of training images and reasonable performance on unseen classes. This is achieved by seeing the distributions of words in texts as a semantic space for understanding what objects look like. Our deep learning model does not require any manually defined semantic or visual features for either words or images. Images are mapped to be close to semantic word vectors corresponding to their classes, and the resulting image embeddings can be used to distinguish whether an image is of a seen or unseen class. We then use novelty detection methods to differentiate unseen classes from seen classes. We demonstrate two novelty detection strategies; the first gives high accuracy on unseen classes, while the second is conservative in its prediction of novelty and keeps the seen classes' accuracy high.

## 1 Introduction

The ability to classify instances of an unseen visual class, called zero-shot learning, is useful in several situations. There are many species and products without labeled data and new visual categories, such as the latest gadgets or car models, that are introduced frequently. In this work, we show how to make use of the vast amount of knowledge about the visual world available in natural language to classify unseen objects. We attempt to model people's ability to identify unseen objects even if the only knowledge about that object came from reading about it. For instance, after reading the description of *a two-wheeled self-balancing electric vehicle, controlled by a stick, with which you can move around while standing on top of it*, many would be able to identify a *Segway*, possibly after being briefly perplexed because the new object looks different from previously observed classes.

We introduce a zero-shot model that can predict both seen and unseen classes. For instance, without ever seeing a cat image, it can determine whether an image shows a cat or a known category from the training set such as a dog or a horse. The model is based on two main ideas.

Fig. 1 illustrates the model. First, images are mapped into a semantic space of words that is learned by a neural network model [15]. Word vectors capture distributional similarities from a large, unsupervised text corpus. By learning an image mapping into this space, the word vectors get implicitly grounded by the visual modality, allowing us to give prototypical instances for various words. Second, because classifiers prefer to assign test images into classes for which they have seen training examples, the model incorporates *novelty detection* which determines whether a new image is on the manifold of known categories. If the image is of a known category, a standard classifier can be used. Otherwise, images are assigned to a class based on the likelihood of being an unseen category. We explore two strategies for novelty detection, both of which are based on ideas from outlier detection methods. The first strategy prefers high accuracy for unseen classes, the second for seen classes.

Unlike previous work on zero-shot learning which can only predict intermediate features or differentiate between various zero-shot classes [21, 27], our joint model can achieve both state of the art accuracy on known classes as well as reasonable performance on unseen classes. Furthermore, compared to related work on knowledge transfer [21, 28] we do not require manually defined semantic

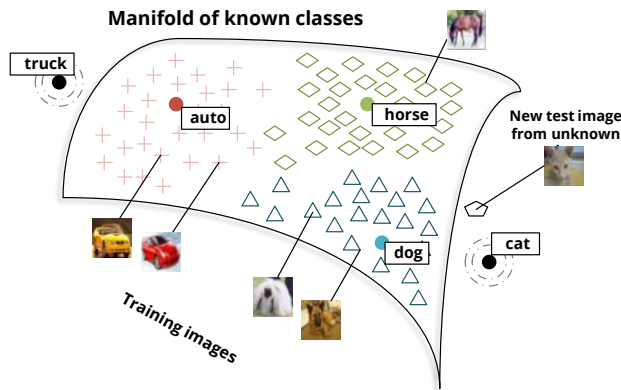

Figure 1: Overview of our cross-modal zero-shot model. We first map each new testing image into a lower dimensional semantic word vector space. Then, we determine whether it is on the manifold of seen images. If the image is 'novel', meaning not on the manifold, we classify it with the help of unsupervised semantic word vectors. In this example, the unseen classes are truck and cat.

or visual attributes for the zero-shot classes, allowing us to use state-of-the-art unsupervised and unaligned image features instead along with unsupervised and unaligned language corpora.

## 2 Related Work

We briefly outline connections and differences to five related lines of research. Due to space constraints, we cannot do justice to the complete literature.

**Zero-Shot Learning.** The work most similar to ours is that by Palatucci et al. [27]. They map fMRI scans of people thinking about certain words into a space of manually designed features and then classify using these features. They are able to predict semantic features even for words for which they have not seen scans and experiment with differentiating between several zero-shot classes. However, they do not classify new test instances into both seen and unseen classes. We extend their approach to allow for this setup using novelty detection. Lampert et al. [21] construct a set of binary attributes for the image classes that convey various visual characteristics, such as "furry" and "paws" for bears and "wings" and "flies" for birds. Later, in section 6.4, we compare our method to their method of performing Direct Attribute Prediction (DAP).

**One-Shot Learning** One-shot learning [19, 20] seeks to learn a visual object class by using very few training examples. This is usually achieved by either sharing of feature representations [2], model parameters [12] or via similar context [14]. A recent related work on one-shot learning is that of Salakhutdinov et al. [29]. Similar to their work, our model is based on using deep learning techniques to learn low-level image features followed by a probabilistic model to transfer knowledge, with the added advantage of needing no training data due to the cross-modal knowledge transfer from natural language.

**Knowledge and Visual Attribute Transfer.** Lampert et al. and Farhadi et al. [21, 10] were two of the first to use well-designed visual attributes of unseen classes to classify them. This is different to our setting since we only have distributional features of words learned from unsupervised, non-parallel corpora and can classify between categories that have thousands or zero training images. Qi et al. [28] learn when to transfer knowledge from one category to another for each instance.

**Domain Adaptation.** Domain adaptation is useful in situations in which there is a lot of training data in one domain but little to none in another. For instance, in sentiment analysis one could train a classifier for movie reviews and then adapt from that domain to book reviews [4, 13]. While related, this line of work is different since there is data for each *class* but the features may differ between domains.

**Multimodal Embeddings.** Multimodal embeddings relate information from multiple sources such as sound and video [25] or images and text. Socher et al. [31] project words and image regions into a common space using kernelized canonical correlation analysis to obtain state of the art performance in annotation and segmentation. Similar to our work, they use unsupervised large text corpora to

learn semantic word representations. Their model does require a small amount of training data however for each class. Some work has been done on multimodal distributional methods [11, 23]. Most recently, Bruni et al. [5] worked on perceptually grounding word meaning and showed that joint models are better able to predict the color of concrete objects.

## 3 Word and Image Representations

We begin the description of the full framework with the feature representations of words and images. Distributional approaches are very common for capturing semantic similarity between words. In these approaches, words are represented as vectors of distributional characteristics – most often their co-occurrences with words in context [26, 9, 1, 32]. These representations have proven very effective in natural language processing tasks such as sense disambiguation [30], thesaurus extraction [24, 8] and cognitive modeling [22].

Unless otherwise mentioned, all word vectors are initialized with pre-trained $d = 50$-dimensional word vectors from the unsupervised model of Huang et al. [15]. Using free Wikipedia text, their model learns word vectors by predicting how likely it is for each word to occur in its context. Their model uses both local context in the window around each word and global document contex, thus capturing distributional syntactic and semantic information. For further details and evaluations of these embeddings, see [3, 7].

We use the unsupervised method of Coates et al. [6] to extract $I$ image features from raw pixels in an unsupervised fashion. Each image is henceforth represented by a vector $x \in \mathbb{R}^I$.

## 4 Projecting Images into Semantic Word Spaces

In order to learn semantic relationships and class membership of images we project the image feature vectors into the $d$-dimensional, semantic word space $F$. During training and testing, we consider a set of classes $Y$. Some of the classes $y$ in this set will have available training data, others will be zero-shot classes without any training data. We define the former as the seen classes $Y_s$ and the latter as the unseen classes $Y_u$. Let $W = W_s \cup W_u$ be the set of word vectors in $\mathbb{R}^d$ for both seen and unseen visual classes, respectively.

All training images $x^{(i)} \in X_y$ of a seen class $y \in Y_s$ are mapped to the word vector $w_y$ corresponding to the class name. To train this mapping, we train a neural network to minimize the following objective function :

$$J(\Theta) = \sum_{y \in Y_s} \sum_{x^{(i)} \in X_y} \left|\left| w_y - \theta^{(2)} f\left( \theta^{(1)} x^{(i)} \right) \right|\right|^2, \tag{1}$$

where $\theta^{(1)} \in \mathbb{R}^{h \times I}$, $\theta^{(2)} \in \mathbb{R}^{d \times h}$ and the standard nonlinearity $f = \tanh$. We define $\Theta = (\theta^{(1)}, \theta^{(2)})$. A two-layer neural network is shown to outperform a single linear mapping in the experiments section below. The cost function is trained with standard backpropagation and L-BFGS. By projecting images into the word vector space, we implicitly extend the semantics with a visual grounding, allowing us to query the space, for instance for prototypical visual instances of a word.

Fig. 2 shows a visualization of the 50-dimensional semantic space with word vectors and images of both seen and unseen classes. The unseen classes are cat and truck. The mapping from 50 to 2 dimensions was done with t-SNE [33]. We can observe that most classes are tightly clustered around their corresponding word vector while the zero-shot classes (cat and truck for this mapping) do not have close-by vectors. However, the images of the two zero-shot classes are close to semantically similar classes (such as in the case of cat, which is close to dog and horse but is far away from car or ship). This observation motivated the idea for first detecting images of unseen classes and then classifying them to the zero-shot word vectors.

## 5 Zero-Shot Learning Model

In this section we first give an overview of our model and then describe each of its components. In general, we want to predict $p(y|x)$, the conditional probability for both seen and unseen classes $y \in Y_s \cup Y_u$ given an image from the test set $x \in X_t$. To achieve this we will employ the semantic vectors to which these images have been mapped to $f \in F_t$.

Because standard classifiers will never predict a class that has no training examples, we introduce a binary *novelty* random variable which indicates whether an image is in a seen or unseen class

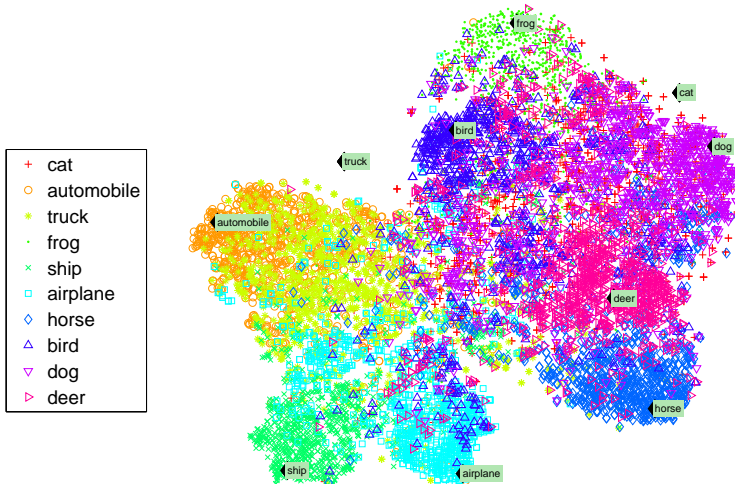

Figure 2: T-SNE visualization of the semantic word space. Word vector locations are highlighted and mapped image locations are shown both for images for which this mapping has been trained and unseen images. The unseen classes are cat and truck.

$V \in \{s, u\}$. Let $X_s$ be the set of all feature vectors for training images of seen classes and $F_s$ their corresponding semantic vectors. We similarly define $F_y$ to be the semantic vectors of class $y$. We predict a class $y$ for a new input image $x$ and its mapped semantic vector $f$ via:

$$p(y|x, X_s, F_s, W, \theta) = \sum_{V \in \{s,u\}} P(y|V, x, X_s, F_s, W, \theta) P(V|x, X_s, F_s, W, \theta).$$

Marginalizing out the novelty variable $V$ allows us to first distinguish between seen and unseen classes. Each type of image can then be classified differently. The seen image classifier can be a state of the art $\mathrm{softmax}$ classifier while the unseen classifier can be a simple Gaussian discriminator.

## 5.1    Strategies for Novelty Detection

We now consider two strategies for predicting whether an image is of a seen or unseen class. The term $P(V = u|x, X_s, F_s, W, \theta)$ is the probability of an image being in an unseen class. An image from an unseen class will not be very close to the existing training images but will still be roughly in the same semantic region. For instance, cat images are closest to dogs even though they are not as close to the dog word vector as most dog images are. Hence, at test time, we can use outlier detection methods to determine whether an image is in a seen or unseen class.

We compare two strategies for outlier detection. Both are computed on the manifold of training images that were mapped to the semantic word space. The first method is relatively liberal in its assessment of novelty. It uses simple thresholds on the marginals assigned to each image under isometric, class-specific Gaussians. The mapped points of seen classes are used to obtain this marginal. For each seen class $y \in Y_s$, we compute $P(x|X_y, w_y, F_y, \theta) = P(f|F_y, w_y) = \mathcal{N}(f|w_y, \Sigma_y)$. The Gaussian of each class is parameterized by the corresponding semantic word vector $w_y$ for its mean and a covariance matrix $\Sigma_y$ that is estimated from all the mapped training points with that label. We restrict the Gaussians to be isometric to prevent overfitting. For a new image $x$, the outlier detector then becomes the indicator function that is $1$ if the marginal probability is below a certain threshold $T_y$ for all the classes:

$$P(V = u|f, X_s, W, \theta) := \mathbb{1}\{\forall y \in Y_s : P(f|F_y, w_y) < T_y\}$$

We provide an experimental analysis for various thresholds $T$ below. The thresholds are selected to make at least some fraction of the vectors from training images above threshold, that is, to be classified as a seen class. Intuitively, smaller thresholds result in fewer images being labeled as unseen. The main drawback of this method is that it does not give a real probability for an outlier.

An alternative would be to use the method of [17] to obtain an actual outlier probability in an unsupervised way. Then, we can obtain the conditional class probability using a weighted combination of classifiers for both seen and unseen classes (described below). Fig. 2 shows that many unseen images are not technically outliers of the complete data manifold. Hence this method is very conservative in its assignment of novelty and therefore preserves high accuracy for seen classes.

We need to slightly modify the original approach since we distinguish between training and test sets. We do not want to use the set of all test images since they would then not be considered outliers anymore. The modified version has the same two parameters: $k = 20$, the number of nearest neighbors that are considered to determine whether a point is an outlier and $\lambda = 3$, which can be roughly seen as a multiplier on the standard deviation. The larger it is, the more a point has to deviate from the mean in order to be considered an outlier.

For each point $f \in F_t$, we define a context set $C(f) \subseteq F_s$ of $k$ nearest neighbors in the *training set* of seen categories. We can compute the *probabilistic set distance* pdist of each point $x$ to the points in $C(f)$:

$$\text{pdist}_\lambda(f, C(f)) = \lambda \sqrt{\frac{\sum_{q \in C(f)} d(f, q)^2}{|C(f)|}},$$

where $d(f, q)$ defines some distance function in the word space. We use Euclidean distances. Next we define the *local outlier factor*:

$$\text{lof}_\lambda(f) = \frac{\text{pdist}_\lambda(f, C(f))}{\mathbb{E}_{q \sim C(f)}[\text{pdist}_\lambda(f, C(q))]} - 1.$$

Large lof values indicate increasing outlierness. In order to obtain a probability, we next define a normalization factor $Z$ that can be seen as a kind of standard deviation of lof values in the *training set* of seen classes:

$$Z_\lambda(F_s) = \lambda \sqrt{\mathbb{E}_{q \sim F_s}[(\text{lof}(q))^2]}.$$

Now, we can define the *Local Outlier Probability*:

$$LoOP(f) = \max \left\{ 0, \text{erf}\left(\frac{\text{lof}_\lambda(f)}{Z_\lambda(F_s)}\right) \right\}, \tag{2}$$

where erf is the Gauss Error function. This probability can now be used to weigh the seen and unseen classifiers by the appropriate amount given our belief about the outlierness of a new test image.

## 5.2 Classification

In the case where $V = s$, i.e. the point is considered to be of a known class, we can use any probabilistic classifier for obtaining $P(y|V = s, x, X_s)$. We use a $\text{softmax}$ classifier on the original $I$-dimensional features. For the zero-shot case where $V = u$ we assume an isometric Gaussian distribution around each of the novel class word vectors and assign classes based on their likelihood.

# 6 Experiments

For most of our experiments we utilize the CIFAR-10 dataset [18]. The dataset has 10 classes, each with 5,000 $32 \times 32 \times 3$ RGB images. We use the unsupervised feature extraction method of Coates and Ng [6] to obtain a 12,800-dimensional feature vector for each image. For word vectors, we use a set of 50-dimensional word vectors from the Huang dataset [15] that correspond to each CIFAR category. During training, we omit two of the 10 classes and reserve them for zero-shot analysis. The remaining categories are used for training.

In this section we first analyze the classification performance for seen classes and unseen classes separately. Then, we combine images from the two types of classes, and discuss the trade-offs involved in our two unseen class detection strategies. Next, the overall performance of the entire classification pipeline is summarized and compared to another popular approach by Lampert et al. [21]. Finally, we run a few additional experiments to assess quality and robustness of our model.

## 6.1 Seen and Unseen Classes Separately

First, we evaluate the classification accuracy when presented only with images from classes that have been used in training. We train a $\text{softmax}$ classifier to label one of 8 classes from CIFAR-10 (2 are reserved for zero-shot learning). In this case, we achieve an accuracy of 82.5% on the set of

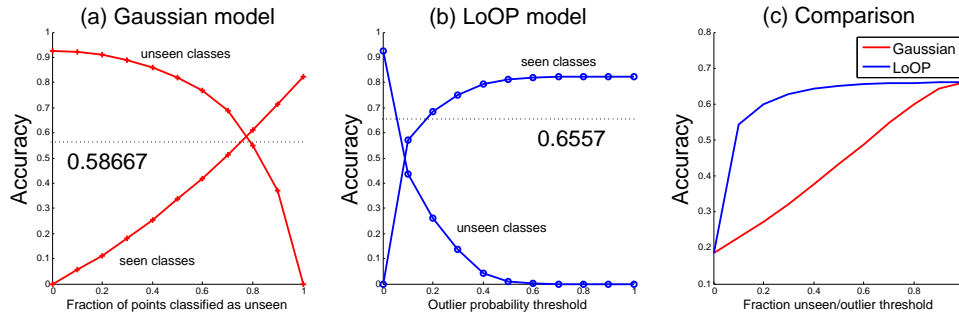

Figure 4: Comparison of accuracies for images from previously seen and unseen categories when unseen images are detected under the (a) Gaussian threshold model, (b) LoOP model. The average accuracy on all images is shown in (c) for both models. We also show a line corresponding to the single accuracy achieved in the Bayesian pipeline. In these examples, the zero-shot categories are "cat" and "truck".

classes excluding cat and truck, which closely matches the SVM-based classification results in the original Coates and Ng paper [6] that used all 10 classes.

We now focus on classification between only two zero-shot classes. In this case, the classification is based on isometric Gaussians which amounts to simply comparing distances between word vectors of unseen classes and an image mapped into semantic space. In this case, the performance is good if there is at least one seen class similar to the zero-shot class. For instance, when *cat* and *dog* are taken out from training, the resulting zero-shot classification does not work well because none of the other 8 categories is similar enough to both images to learn a good semantic distinction. On the other hand, if *cat* and *truck* are taken out, then the cat vectors can be mapped to the word space thanks to similarities to *dogs* and *trucks* can be distinguished thanks to *car*, yielding better performance.

Fig. 3 shows the accuracy achieved in distinguishing images belonging to various combinations of zero-shot classes. We observe, as expected, that the maximum accuracy is achieved when choosing semantically distinct categories. For instance, frog-truck and cat-truck do very well. The worst accuracy is obtained when cat and dog are chosen instead. From the figure we see that for certain combinations of zero-shot classes, we can achieve accuracies up to 90%.

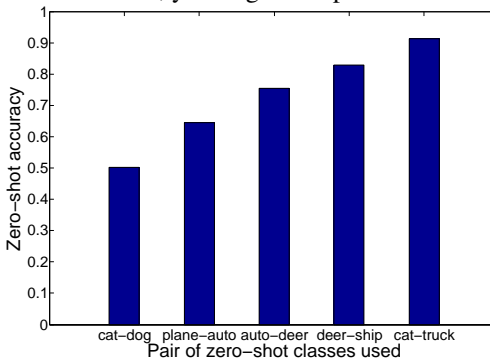

## 6.2 Influence of Novelty Detectors on Average Accuracy

Our next area of investigation is to determine the average performance of the classifier for the overall dataset that includes both seen and unseen images. We compare the performance

Figure 3: Visualization of classification accuracy achieved for unseen images, for different choices of zero-shot classes selected before training.

when each image is passed through either of the two novelty detectors which decide with a certain probability (in the second scenario) whether an image belongs to a class that was used in training. Depending on this choice, the image is either passed through the softmax classifier for seen category images, or assigned to the class of the nearest semantic word vector for unseen category images.

Fig. 4 shows the accuracies for test images for different choices made by the two scenarios for novelty detection. The test set includes an equal number of images from each category, with 8 categories having been seen before, and 2 being new. We plot the accuracies of the two types of images separately for comparison. Firstly, at the left extreme of the curve, the Gaussian unseen image detector treats all of the images as unseen, and the LoOP model takes the probability threshold for an image being unseen to be 0. At this point, with all unseen images in the test set being treated as such, we achieve the highest accuracies, at 90% for this zero-shot pair. Similarly, at the other extreme of the curve, all images are classified as belonging to a seen category, and hence the softmax classifier for seen images gives the best possible accuracy for these images.

Between the extremes, the curves for unseen image accuracies and seen image accuracies fall and rise at different rates. Since the Gaussian model is liberal in designating an image as belonging to an unseen category, it treats more of the images as unseen, and hence we continue to get high unseen class accuracies along the curve. The LoOP model, which tries to detect whether an image could be regarded as an outlier for each class, does not assign very high outlier probabilities to zero-shot images due to a large number of them being spread on inside the manifold of seen images (see Fig. 2 for a 2-dimensional visualization of the originally 50-dimensional space). Thus, it continues to treat the majority of images as seen, leading to high seen class accuracies. Hence, the LoOP model can be used in scenarios where one does not want to degrade the high performance on classes from the training set but allow for the possibility of unseen classes.

We also see from Fig. 4 (c) that since most images in the test set belong to previously seen categories, the LoOP model, which is conservative in assigning the unseen label, gives better overall accuracies than the Gaussian model. In general, we can choose an acceptable threshold for seen class accuracy and achieve a corresponding unseen class accuracy. For example, at 70% seen class accuracy in the Gaussian model, unseen classes can be classified with accuracies of between 30% to 15%, depending on the class. Random chance is 10%.

### 6.3 Combining predictions for seen and unseen classes

The final step in our experiments is to perform the full Bayesian pipeline as defined by Equation 2. We obtain a prior probability of an image being an outlier. The LoOP model outputs a probability for the image instance being an outlier, which we use directly. For the Gaussian threshold model, we tune a cutoff fraction for log probabilities beyond which images are classified as outliers. We assign probabilities 0 and 1 to either side of this threshold. We show the horizontal lines corresponding to the overall accuracy for the Bayesian pipeline on Figure 4.

### 6.4 Comparison to attribute-based classification

To establish a context for comparing our model performance, we also run the attribute-based classification approach outlined by Lampert et al. [21]. We construct an attribute set of 25 attributes highlighting different aspects of the CIFAR-10 dataset, with certain aspects dealing with animal-based attributes, and others dealing with vehicle-based attributes. We train each binary attribute classifier separately, and use the trained classifiers to construct attribute labels for unseen classes. Finally, we use MAP prediction to determine the final output class. The table below shows a summary of results. Our overall accuracies for both models outperform the attribute-based model.

| | |
|---|---|
| Bayesian pipeline (Gaussian) | 74.25% |
| Bayesian pipeline (LoOP) | 65.31% |
| Attribute-based (Lampert et al.) | 45.25% |

In general, an advantage of our approach is the ability to adapt to a domain quickly, which is difficult in the case of the attribute-based model, since appropriate attribute types need to be carefully picked.

### 6.5 Novelty detection in original feature space

The analysis of novelty detectors in 6.2 involves calculation in the word space. As a comparison, we perform the same experiments with the Gaussian model in the original feature space. In the mapped space, we observe that of the 100 images assigned the highest probability of being an outlier, 12% of those images are false positives. On the other hand, in the original feature space, the false positive rate increases to 78%. This is intuitively explained by the fact that the mapping function gathers extra semantic information from the word vectors it is trained on, and images are able to cluster better around these assumed Gaussian centroids. In the original space, there is no semantic information, and the Gaussian centroids need to be inferred from among the images themselves, which are not truly representative of the center of the image space for their classes.

### 6.6 Extension to
### CIFAR-100 and Analysis of Deep Semantic Mapping

So far, our tests were on the CIFAR-10 dataset. We now describe results on the more challenging CIFAR-100

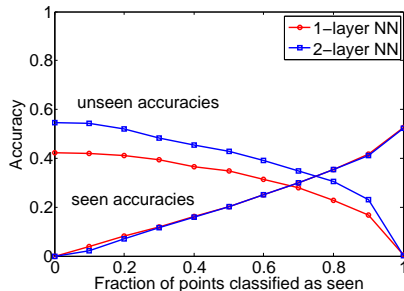

Figure 5: Comparison of accuracies for images from previously seen and unseen categories for the modified CIFAR-100 dataset, after training the semantic mapping with a one-layer network and two-layer network. The deeper mapping function performs better.

dataset [18], which consists of 100 classes, with 500 $32 \times 32 \times 3$ RGB images in each class. We remove 4 categories for which no vector representations were available in our vocabulary. We then combined the CIFAR-10 dataset to get a set of 106 classes. Six zero-shot classes were chosen: 'forest', 'lobster', 'orange', 'boy', 'truck', and 'cat'. As before, we train a neural network to map the vectors into semantic space. With this setup, we get a peak non-zero-shot accuracy of 52.7%, which is almost near the baseline on 100 classes [16]. When all images are labeled as zero shot, the peak accuracy for the 6 unseen classes is 52.7%, where chance would be at 16.6%.

Because of the large semantic space corresponding to 100 classes, the proximity of an image to its appropriate class vector is dependent on the quality of the mapping into semantic space. We hypothesize that in this scenario a two layer neural network as described in Sec. 4 will perform better than a single layer or linear mapping. Fig. 5 confirms this hypothesis. The zero-shot accuracy is 10% higher with a 2 layer neural net compared to a single layer with 42.2%.

### 6.7   Zero-Shot Classes with Distractor Words

We would like zero-shot images to be classified correctly when there are a large number of unseen categories to choose from. To evaluate such a setting with many possible but incorrect unseen classes we create a set of distractor words. We compare two scenarios. In the first, we add random nouns to the semantic space. In the second, much harder, setting we add the $k$ nearest neighbors of a word vector. We then evaluate classification accuracy with each new set. For the zero-shot class *cat* and *truck*, the nearest neighbors distractors include *rabbit*, *kitten* and *mouse*, among others.

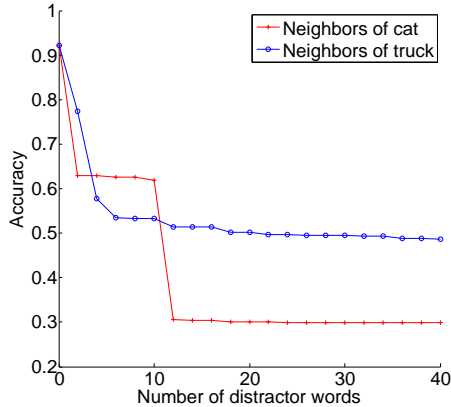

Figure 6: Visualization of the zero-shot classification accuracy when distractor words from the nearest neighbor set of a given category are also present.

The accuracy does not change much if random distractor nouns are added. This shows that the semantic space is spanned well and our zero-shot learning model is quite robust. Fig. 6 shows the classification accuracies for the second scenario. Here, accuracy drops as an increasing number of semantically related nearest neighbors are added to the distractor set. This is to be expected because there are not enough related categories to accurately distinguish very similar categories. After a certain number, the effect of a new distractor word is small. This is consistent with our expectation that a certain number of closely-related semantic neighbors would distract the classifier; however, beyond that limited set, other categories would be further away in semantic space and would not affect classification accuracy.

## 7   Conclusion

We introduced a novel model for jointly doing standard and zero-shot classification based on deep learned word and image representations. The two key ideas are that (i) using semantic word vector representations can help to transfer knowledge between modalities even when these representations are learned in an unsupervised way and (ii) that our Bayesian framework that first differentiates novel unseen classes from points on the semantic manifold of trained classes can help to combine both zero-shot and seen classification into one framework. If the task was only to differentiate between various zero-shot classes we could obtain accuracies of up to 90% with a fully unsupervised model.

#### Acknowledgments

Richard is partly supported by a Microsoft Research PhD fellowship. The authors gratefully acknowledge the support of the Defense Advanced Research Projects Agency (DARPA) Deep Exploration and Filtering of Text (DEFT) Program under Air Force Research Laboratory (AFRL) prime contract no. FA8750-13-2-0040, the DARPA Deep Learning program under contract number FA8650-10-C-7020 and NSF IIS-1159679. Any opinions, findings, and conclusions or recommendations expressed in this material are those of the authors and do not necessarily reflect the view of DARPA, AFRL, or the US government.

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
