[Reviews · NeurIPS 2013]

Submitted by Assigned_Reviewer_4

This paper proposes a model for labeling images with classes for which no example appear in the training set, which is based on a combination of word and image embeddings and novelty detection. Using distances in th embedding space between test images and unseen and seen class labels, the approach is able to assign a probability for a new image to be from an unseen class. This is later used to decide which classifier to use (one designed for seen classes the other for unknown ones). Results on CIFAR10 are provided.



Clarity: The paper is clearly written. Its motivation and studied task are interesting and ambitious.

Originality: There is not much technical novelty since image and word embedding methods come from previous work and so do novelty detections models (with a slight modification of [Kriegel et al. CIKM09]). Still, the combination of those approach appears to be brand new and is not trivial.

Significance: The importance of the problem considered here make that this paper could have quite an impact. However, the true application setting of such work (large numbers of classes, imbalance) is still quite far from what is proposed in this paper, which is yet an interesting step towards it. Indeed, all experiments here considers balanced classes, and it seems that such embedding-based work might suffer with imbalance (and the isometric Gaussian assumption), especially when the majority are examples from unseen classes. Similarly, this system needs "well-chosen" unseen classes to perform well (as shown in Fig. 3, where the system performs as chance when unseen classes are cat and dog). Still, these difficulties are shared by any existing system that I know of.

Quality: The overall quality of the work is good but some questions remain. The more important point is in Section 6.3, where the full pipeline is used. This shows that in this setting, the Gaussian model outperforms the more sophisticated LoOP approach. Could it be due to the cutoff fraction for log probabilities, which is well tuned for the Gaussian model? How has it been chosen? On the training set? The LoOP does not require (nor benefit from) such tuning. What is the proportion of test examples set as unseen and seen in this combined setting?

In Section 6.6, it is not specified if LoOP or the Gaussian model was used. According to the shape of curve of Fig. 5, I would say the Gaussian model. Is it true? Does it mean that this simpler approach is actually the one to choose for good performances with increased number of classes?

Other remarks/questions:
- Are the word vectors $w_y$ updated?
- There seems to be an error in the equation defining lof(.): in the expectation (denominator), shouldn't we read pdist_\lambda(q,C(q)) ? (q instead of f).
- A legend should be added for Figure 4 (left and center) indicating which curve correspond to accuracy on seen class, and to unseen.
- Comparison with (Lampert et al., 09) in Section 6.3 is interesting. One should also compare with "What Helps Where -- And Why? Semantic Relatedness for Knowledge Transfer" by Rohrbach et al. CVPR 2010, which improved over Lampert and proposes an approach that does not rely on attributes for zero-shot learning.
- The results with random distractor noise should be added in Figure 6.

After rebuttal: I read the rebuttal which answered conveniently some questions I had. But does not completely justify the novelty.
Summary: This paper tackles the problem of classifying images for which no labeled sample was provided in training. The approach, based on embedding works and outlier detection, is original and promising, even if the experimental setting remains simple compared to real application settings and that some precisions should be given.

Submitted by Assigned_Reviewer_5

This paper presents a method to solve the zero-shot learning of unseen image classes. An image is represented using two feature vectors one containing image related features whereas the other contains natural language texts (words) as features. A manifold is then learnt such that a non-linear transformation of the lexical feature vector and the image feature vector is projected closer to each other (in L2 sense) in this manifold. Very encouraging results are obtained on two standard image classification dataset for unseen image categories.

The paper is well-written and easy to read. The proposed method is analyzed from different aspects such as the effect of the distractors. Although some of the image classes are hand picked for evaluation and visualization purposes this does not affect the general applicability of the proposed method.

Two things that came to my mind while reviewing the paper are as follows:

a) The non-linear transformation (tanh) happens on the image feature vector side. Have you tried the other way around? Is there any reason to specifically select the image feature vector and not the image feature vector as the input to the neural network (on which the non-linear transformation is carried upon)? If the objective was the produce text for an image (image annotation) then obviously this setting would be the natural one. But from the point of view of learning a manifold that embeds image and lexical feature vectors close together, I think the decision is symmetric. From a very general perspective, the task can be seen as performing CCA between image and lexical features (perhaps a kernalized version of CCA if you want to introduce non-linearity into the mapping). In this regard, both image as well as lexical features could have been non-linearly transformed first and then found a mapping between those transformed vectors. Perhaps these alternative were tried but without any success?

b) It would have been interesting if the authors presented the predicted lexical features for some unseen images. For example, given an image of a cat, does the word cat appear among the highest weighted features?
Summary: This paper presents a method to solve the zero-shot learning of unseen image classes. An image is represented using two feature vectors one containing image related features whereas the other contains natural language texts (words) as features. A manifold is then learnt such that a non-linear transformation of the lexical feature vector and the image feature vector is projected closer to each other (in L2 sense) in this manifold. Very encouraging results are obtained on two standard image classification dataset for unseen image categories.

Submitted by Assigned_Reviewer_6

This paper deals with the problem of categorizing images where some classes of images have been seen before and others have not. The task is to identify the class for both those that have been seen before and for those that have not been seen before. This is done by mapping the images to a 50D semantic space obtained from the category labels based on their word co-occurrence statistics.


Novelty

The general idea of using another view of the data to help warp the space for one view of the data (e.g. vision) in unsupervised learning has been around for a long time (see for example Carpenter, Grossberg, & Reynolds 91, Becker & Hinton 92, de Sa 94, Schmidhuber and Prelinger 93, Phillips et al. 95, and Kay et al. 98). The application
to zero-shot learning was done by Palatucci et al (ref 27 in the paper). Here the
authors have added the ability to also classify the previously experienced classes as well as the previously unseen classes.

Clarity

The paper is fairly easy to read and understand, however (very minor point) it is not clear to me why a 2 layer NN is called a deep network. Figure 2 clearly illustrates the issues involved.


Quality

The comparison to doing the novelty detection in the original space is a nice and important control. Comparing to the Lampert et al. model is also nice, though without knowing what attributes were picked it is hard to evaluate how fair this comparison is.

The experiments are well run and the different choices of left-out classes clearly illustrate the issues involved and the dependence of the results on similarity of the left-out items to the experienced items.

Summary: This paper uses a semantic mapping to peform zero-shot learning on unseen classes. The novelty over prior approaches is that seen classes can also be classified. The work is well done and clearly written but seems more incremental than ground-breaking.
Author Feedback

Author rebuttal: We thank the reviewers for their insightful comments.

@Reviewer_4
About Section 6.3: Yes, we gave the simpler Gaussian baseline model the big (unfair) advantage of tuning the threshold of the log probability on a dev set.
This is fixed in the camera ready by using the other seen classes to find the optimal threshold.

About Section 6.6: Yes, thanks for catching this. This was only the Gaussian model since it’s simpler and allows to see the impact of the neural net better but we will add the LoOp model there as well.

Are the word vectors $w_y$ updated?
No, because that would break the system for unseen words since they will stay in their original space but the seen class words will move around.

@Reviewer_5
a) The non-linear transformation: These are interesting considerations. We tried applying the tanh to the word vectors also but it did not improve performance.

b) That is a great idea! We will provide some examples and show the nearest neighboring words for several images so that we can possibly see some more dog-like cat images and other interesting phenomena that the model captures. Thank you!

@Reviewer_6
Thank you for the additional references that we will include.
We will also include the list of attributes which we carefully chose to be reasonable for the image classes.